# Feasibility of a Geriatric Assessment to Detect and Quantify Sarcopenia and Physical Functioning in German Nursing Home Residents—A Pilot Study

**DOI:** 10.3390/geriatrics6030069

**Published:** 2021-07-02

**Authors:** Daniel Haigis, Rebekka Pomiersky, Dorotheé Altmeier, Annika Frahsa, Gorden Sudeck, Ansgar Thiel, Gerhard Eschweiler, Andreas Michael Nieß

**Affiliations:** 1Department of Sports Medicine, University Hospital of Tuebingen, 72076 Tübingen, Germany; andreas.niess@med.uni-tuebingen.de; 2Interfaculty Research Institute for Sport and Physical Activity, University of Tuebingen, 72074 Tübingen, Germany; rebekka.pomiersky@uni-tuebingen.de (R.P.); dorothee.altmeier@uni-tuebingen.de (D.A.); annika.frahsa@uni-tuebingen.de (A.F.); gorden.sudeck@uni-tuebingen.de (G.S.); a.thiel@uni-tuebingen.de (A.T.); 3Institute of Sport Science, Eberhard Karls University of Tuebingen, 72074 Tübingen, Germany; 4Institute of Social and Preventive Medicine, University of Bern, 3012 Bern, Switzerland; 5Centre for Geriatric Medicine, University Hospital of Tuebingen, 72076 Tübingen, Germany; gerhard.eschweiler@med.uni-tuebingen.de

**Keywords:** sarcopenia, prevalence, nursing home, physical functioning

## Abstract

Background: Entering into a nursing home leads to increased immobility and further reductions in physical and cognitive functioning. As a result, there is a risk of sarcopenia, which is characterized by loss of muscle strength, muscle mass and physical functioning. To our knowledge, the feasibility of sarcopenia screening has not yet been performed in the German nursing home setting. Methods: For sarcopenia screening, the specifications of EWGSOP2 were applied. The quantification of sarcopenia was performed according to the corresponding cut-off values. The collection of anthropometric data and the morbidity status were recorded. SARC-F, mini-mental state examination, Barthel Index, Short Physical Performance Battery and Timed Up and Go tests were implemented. Results: In one participant, severe sarcopenia could be identified. The quantification was not possible for four participants. A suspicion of sarcopenia was not confirmed in five participants. Only one person was able to perform all assessments. Conclusions: Sarcopenia screening according to EWGSOP2 presented satisfactory feasibility by nursing home residents. However, further tests to assess the physical functioning of the participants often could not be performed. Moreover, inconsistencies in individual assessments became apparent, leading to inconclusive analyses. The recording of sarcopenia prevalence in German nursing homes should be the goal of further research.

## 1. Introduction

The population in Germany shows a rapid increase in the age group of individuals 80 years and older. By the year 2050, the percentage of 80-year-olds will rise to 13% of the total population. This growth of 5.2% from the year 2020 (7.2%) shows that demographic change will be one of the major challenges facing the healthcare system. Simultaneously, the need of care and the need of associated support in activities of daily living between the ages of 60 and 80 increases by about 7%. By comparison, the probability of needing care from the age of 80 onwards is around 37%. Predominant factors for this increasing probability are the increase in multimorbidity, as well as physical and cognitive impairments with rising age [1]. The result of this trend is a growing demand for nursing services by relatives and nursing professionals. In this context, inpatient care facilities, such as nursing homes, offer a possibility of everyday support [2]. Entering into a nursing home poses the risk of increasing immobility [3], which promotes pain, fear of falling and reduced physical and cognitive performance for the residents [4]. This is indicated by the loss of (I) muscle strength and (II) muscle mass, as well as (III) physical functioning. Age-related loss of muscle strength and muscle mass is defined as sarcopenia. If there is a reduction in muscle strength and muscle quantity or quality, sarcopenia is considered confirmed. The diagnosis of severe sarcopenia exists when all three characteristics are present [5]. Sarcopenia is a predictor of all-cause mortality in nursing home residents [6]. Monitoring of a developing decrease in muscle mass and physical function could help to initiate appropriate and specific interventions in time to prevent or slow down the onset of sarcopenia. The European Working Group of Sarcopenia on Older People 2 (EWGSOP2) has developed a diagnostic algorithm for the quantification of sarcopenia. The algorithm is based upon standardized test procedures that are already frequently used in clinical and rehabilitative settings [7]. In 2010, cut-off values were already specified by EWGSOP [8]. Some studies have since been established in European countries [9,10,11,12,13] and have shown wide dispersion of sarcopenia prevalence, due to heterogeneity of the studies [14]. However, it remains unclear how to implement the test procedures recommended by EWGSOP2 and how to quantify sarcopenia in nursing home residents. The BaSAlt study project (Verhältnisorientierte Bewegungsförderung und individuelle Bewegungsberatung im Setting Altenwohnheim‘-ein biopsychosoziales Analyse-und Beratungsprojekt) is one of the first to investigate the prevalence of sarcopenia in the German nursing home setting. Initial findings and expectations will be presented by this pilot study regarding the feasibility of sarcopenia screening in the setting of daily routine in a nursing home and its quantification according to EWGSOP2.

## 2. Materials and Methods

### 2.1. Study Design and Subjects

Descriptive data analysis was performed with basic data of the cohort to determine the prevalence of sarcopenia in a nursing home setting. The mean values with standard deviations (mean ± SD), median values (Md) and percentages were used to characterize the study group. Spearman correlation coefficients were additionally calculated for primary outcomes. Correlations were identified as significant with a *p* value ≤0.05. The pilot study was conducted in a nursing home in the district of Reutlingen, Baden-Wuerttemberg (Germany) in March 2020. At the time of recruitment, 46 residents were living in the nursing home. A total of 40 of the individuals had a nursing degree ≤4. The recruitment was carried out by the nursing home manager who presented the study information and obtained written consent from the residents or their authorized representatives. Personal consulting between the study administration and nursing home staff was possible at all times during the study.

### 2.2. Inclusion and Exclusion Criteria

The inclusion criteria for residents were determined by a degree of care ≤4 (classification in German care system in degree of care 1–5) and the ability to walk with or without walking aids. Participation was voluntary and could be interrupted either before or at any time during the assessments. Residents with a degree of care 5 could not be included due to their bedriddenness and lack of other physical or mental abilities. Residents who did not speak German language could not be included either.

### 2.3. Instruments

For sarcopenia screening, standardized test procedures according to EWGSOP2 guidelines were followed [7], as well as other geriatric tests to assess the physical functioning of individuals. EWGSOP2 algorithm is used to quantify and classify sarcopenia into levels of severity. Figure 1 shows the procedure through the different categories. For the geriatric assessment, primary and secondary outcomes were formulated. The collection of basic data served to provide a comprehensive description of the residents. This included age, sex, degree of care, as well as results of the mini-mental state examination (MMSE) [15] and Barthel Index (BI) [16]. Furthermore, morbidities were categorized into (1) past cardiovascular events, (2) arterial hypertension, (3) coronary heart disease, (4) cardiac insufficiency, (5) cardiac pacemaker, (6) post-stroke/cerebral hemorrhage/TIA, (7) chronic lung disease, (8) cancer, (9) diabetes mellitus I or II, (10) osteoarthritis of upper or lower extremities with or without TEP and (11) psychological/emotional/nervous disease of the persons to be examined. Anthropometric data (height in m, weight in kg and BMI in kg/m^2^) were collected for the primary outcomes. The screening questionnaire SARC-F [17] was used to show possible sarcopenia risk through subjective self-assessment of participants. Five questions in different categories (strength, assistive support when walking, getting up from a chair, climbing stairs and falls) were determined based on a point score. Here, a total of 0–10 points could be achieved, with 0–2 points awarded in each category. A point score ≥4 points in total sum was interpreted as a predictor of sarcopenia [7]. For objective sarcopenia identification and quantification [7], hand force measurement [18], bioimpedance analysis [19] and walking speed [20] were performed. By hand force measurement (HFM in kg), two measurements were taken alternately on the right and left hand with an isometric hand force dynamometer (Hydraulic Hand Force Dynamometer Saehan Model SH5001, Saehan, Changwon-si, South Korea). For the measurement of the appendicular skeletal muscle mass (ASMM in kg) and skeletal muscle mass index (SMI in kg/m^2^), the bioimpedance analyzer by AKERN (impedance vector analyzer BIA 101 BIVA, 50 kHz ± 1% measuring frequency) was used. Analysis of generated data was conducted using the software BodygramPlus Enterprise (Version 1.2.2.9, Akern s.r.l., Pontassieve, Italy) for further evaluation. Walking speed was performed with a 4-meter walking speed test (4MWST in m/s) according to the Short Physical Performance Battery (SPPB) specifications. Both habitual and maximum walking speed were recorded over a walking distance of four meters and measured with support of a standardized stopwatch by hand. Before and after measurement, run-on and run-off distances of two meters were taken into account.

Further tests from SPPB were applied for secondary outcomes. A balance test was performed utilizing three different stand variants (parallel stand, semi-tandem stand and tandem stand) [20], with support of a Kistler^®^ force plate (type Z15577, Kistler Instrumente GmbH, Sindelfingen, Germany) for calculating the center of pressure (CoP) in path length (in mm) and velocity of sway (in m/s) [21]. Finally, the 5-Chair-Rise Test (5CRT, in sec) was applied within the SPPB. The cut-off value of sarcopenia screening for a single test was >15 s for five repetitions [7]. The overall score for SPPB was calculated at the completion of the tests. A maximum score of 12 points could be achieved, with a higher score indicating good overall mobility. Aggregate scores of 0–4 points could be considered low, 5–8 points average and 9–12 points high [22]. To conclude secondary testing, three different Timed Up and Go tests (TUG, in sec) were performed. The TUG_single task_ aims at getting up from a chair, walking across a line three meters away, turning around and returning to the starting position on the chair [23]. Another motor test is the TUG_motor imagery_, which corresponds to identical procedure of TUG_single task_, but contains a purely mental concept of the sequence [24]. Lastly, the TUG_dual task_ takes place, which requires additional transport of a filled water glass over the route and specifications of TUG_single task_ [25]. Table 1 shows the collected basic data and individual assessments with corresponding outcomes. The summary of the formulated cut-off values of EWGSOP2 is presented in Table 2.

## 3. Results

### 3.1. Demographic Data

In total, 14 participants were recruited for the study. One resident stated that he had no further interest in participating. Two other residents could not participate due to cognitive disabilities or physical impairments caused by wheelchair immobilization. Another resident was hospitalized during the time of the assessment. In March 2020, 10 participants were thus examined and included in the further evaluation. Table 3 lists the demographic, physical, cognitive and clinical data of the respective study participants.

The study included eight women and two men with an average age of 87.5 years (SD ± 5.8 years) and degree of care of 3 (Md). Average values for MMSE and BI were 22.1 (SD ± 5.9) and 78.8 (SD ± 11.3), respectively. Cognitive impairments limited the execution of MMSE and BI for P8 and P9, while residents 4 and 5 showed MMSE scores <15, which indicates probable dementia. A total of 80% of the participants used a walking aid (all rollators) for the assessments. For the morbidity status, percentage frequencies were shown for the diseases in the categories (1) past cardiovascular event, 30%, (2) arterial hypertension, 70%, (3) coronary heart disease, 30%, (4) cardiac insufficiency, 50%, (5) cardiac pacemakers, 20%, (6) post-stroke/cerebral hemorrhage/TIA, 20%, (7) chronic lung disease, 30%, (8) cancer, 40%, (9) diabetes mellitus II, 20%, (10) osteoarthrosis lower extremity, 40% (50% with/50% without TEP), and (11) psychological/emotional/nervous disease, 40%.

### 3.2. Geriatric Assessment

The results of the individual participants, divided into primary and secondary outcomes, are presented separately in Table 4 and Table 5.

For primary outcomes, BMI was on average 25.9 kg/m^2^ (SD ± 5.8 kg/m^2^), SARC-F cumulative score 5 points (Md), maximum hand force 18.8 kg (SD ± 6.2 kg), ASMM 15.7 kg (SD ± 4.5 kg), SMI 7.2 kg/m^2^ (SD ± 1.5 kg/m^2^), habitual walking speed 0.43 m/s (SD ± 0.16 m/s) and maximum walking speed 0.61 m/s (SD ± 0.09 m/s). The two-sided correlation test by Spearman showed significant correlation for the habitual walking speed with the MMSE (r = 0.717, *p* = 0.045). In addition, significant negative correlation existed between habitual walking speed and maximal hand grip strength (r = 0.778, *p* = 0.008).

For P5, bioimpedance analysis could not be performed for the primary outcomes due to eczema on both feet, making it impossible to place the distal electrodes. For P6 and P8, each reported two failed attempts for the maximum walking speed. Due to external disturbances caused by residents in the hallways of the nursing home, no results could be generated. Based on severe cognitive impairment, SARC-F could not be evaluated in P9 because two questions were not answered. The secondary outcomes could only be fully measured for P4. P1 and P5 completed the TUGsingle task, P1 also completed the TUGmotor imagery. P5 reported two failed attempts on the TUGmotor imagery. For P2, P3, P6, P7, P8, P9 and P10, it was not possible to perform the 5CRT, SPPB, TUG tests, nor the balance measurement on the Kistler^®^ force plate, as a result of their limited physical functioning. Quantification of sarcopenia in the sample was presented in Table 6 according to EWGSOP2 criteria.

After quantification by the algorithm of EWGSOP2, P1 was found to have severe sarcopenia with physical functional limitations. Sarcopenia could not be confirmed in any other case. Concerning P4, it was unclear whether there was a case of reduced muscle strength and physical functional impairment, as there were contradictory results in various tests. The hand force measurement did not fall below the gender-specific cut-off value of <16 kg. However, for the 5CRT, a completed time for five repetitions >15 s was recorded. For walking speed and SPPB, the cut-off values were exceeded (≥8 points), but not for the TUGsingle task (≥20 s). Therefore, a quantification of sarcopenia based on our tests was not possible. Similarly, no conclusion could be made regarding muscle quality for P5, P8 and P9. Additionally, for the muscle quantity in P8 and P9, both ASMM and SMI values reported different results for the cut-off values, thereby the suspected sarcopenia was unable to be confirmed. The cut-off values of the ASMM were not reached. The values of SMI, adjusted for individual body size and skeletal muscle mass, did not indicate the presence of sarcopenia. For P5, BIA measurement could not be performed for medical reasons. In five participants (P2, P3, P6, P7, P10), no sarcopenia was detected. Based on habitual and maximum walking speed, eight of the ten participants showed physical functional impairments.

## 4. Discussion

To the best of our knowledge, the sarcopenia screening, according to EWGSOP2 guidelines, was performed for the first time in a nursing home setting in Germany. However, individual assessments presented difficulties. Possible explanations are cognitive and physical impairments of a multimorbid participant group in our pilot study. A low degree of mobility is required for the assessments, but this is no longer the case for some nursing home residents. Mobilization with a rollator was also ensured in our study collective; however, bedridden residents and wheelchair users could not be considered. Furthermore, the additional presence of a caregiver turned out to be advantageous, because it provided the examiner with expert support for the assessments and assistance with the mobilization of the residents—for example, from bed or chair. The inclusion of an internal caregiver also supported interaction and communication with the participants. Due to the increased risk of falls among the residents, an additional person was added as security. Despite the announcement, study clarification and information during recruitment by the nursing home management and the already signed agreement, only ten of 14 recruited residents could be included in the final study. This showed a further challenge for the data collection of sarcopenia prevalence in our pilot study. Multimorbidity, as well as rapidly changing states of health and motivation of residents, required a high level of flexibility on the part of the examiners. Additionally, this made it considerably more difficult to plan examination periods and slots in the institution.

Certain physical assessments were not possible in this group. Only one participant was able to perform the complete geriatric assessment as planned. Predominantly, individual assessments with the task of getting up from a chair without the help of another person were no longer feasible for most participants. As a result, the TUG tests and the 5CRT could only provide results in some participants. Moreover, balance tests in three stand variants were planned for the SPPB. However, without the safety of holding on to a rollator or the examiner, this could not be performed without reducing the risk of falling during the assessment. There were no adverse events during the study period. Structuring of the geriatric assessment into primary and secondary outcomes in pre-study periods proved to be helpful. By implementing relatively low-risk test procedures for the generation of primary outcomes (SARC-F, hand strength, muscle quality and gait speed), individual assessments were selected which showed satisfactory application and acceptance by the participants. The requirements of EWGSOP2 by testing methods for physical functioning are too demanding for residents in nursing homes. It should therefore be considered whether physical functioning should be required as a relevant test for the quantification of sarcopenia. The need for care of the residents is also shown by the accompanying physical functional limitations. Testing of muscle strength and muscle mass can already identify a confirmed sarcopenia. Severe sarcopenia could be considered if the physical function tests are not feasible.

However, there were discrepancies in the quantification of sarcopenia using the EWGSOP2 algorithm. Thus, the examination of muscle strength and various physical function tests showed partially inconclusive results for the evaluation of participants. Only one person was able to walk faster than the cut-off value over four meters (≤0.8 m/s). Another person achieved divergent results when measuring muscle strength by hand force dynamometer and 5CRT. Additionally, required values could not be recorded in SPPB and walking speed, whereas only the cut-off value of TUG_single task_ was achieved. Hence, it was not clear to what extent there was reduced muscle strength or physical functionality. Consequently, confirmed or severe sarcopenia could not be diagnosed. Furthermore, the muscle quality assessment of the two residents seemed questionable, as the two methods of measuring muscle mass by ASMM and SMI showed different results. To clarify these uncertainties, a differentiated consideration and possible adjustment of the cut-off values should be analyzed and discussed in future studies. Previous scientific studies showed a high prevalence of sarcopenia in different countries and settings, especially among the older population. However, divergent measuring systems and parameters were involved for quantification [14,26]. A study by Papadopoulou and colleagues (2020) compared the prevalence of sarcopenia in three different settings (assisted living, nursing homes and clinical institutions). Nursing homes reported the highest percentage of sarcopenia diagnoses with 31% (95% CI: 22–42%) for women and 51% (95% CI: 37–66%) for men [26]. As a result of the different measurement and quantification methods, cut-off values were controversial and discussed. In a review by Shafiee et al. (2017), a higher prevalence of sarcopenia could be identified in the analysis of muscle quality or muscle mass by BIA measurement, compared to other measurement methods, such as DXA (female: 20% versus 11%; male: 19% versus 10%) [14]. Another study by Masanés and colleagues (2017) showed the influence of the large variability of cut-off values concerning muscle mass. The authors reported that even small deviations of the SMI (women: +1.23 kg/m^2^; men: +1.62 kg/m^2^) demonstrated a strong influence on the sarcopenia diagnosis (women: 1%–22%; men: 6%–41%). Unfortunately, influences of muscle strength and gait speed indicated only a minor impact [27]. Therefore, the measurement of muscle mass should be the focus of future studies on this topic. This is also shown by the present study. Assessments to determine muscle strength using an isometric hand strength test and that of physical functioning considering habitual gait speed were possible for all study participants. In two individuals, a difference between the two survey methods was identified in regard to muscle mass. The resulting sarcopenia diagnoses would differ in one of the individuals. The argument that the cut-off values should be checked is supported by the study by Perkisas et al. (2019). Mainly specific cohorts, such as nursing home residents, should be surveyed, as muscle mass is considered an important predictor of all-cause mortality [28].

The difference in muscle strength testing methods for the upper body and lower body presents another difficulty for quantifying sarcopenia prevalence. For the majority of participants, the completion of the 5CRT was not possible. This could indicate an early reduction of the muscle strength of the lower body, which can lead to a contradictory result of the hand grip strength measurement. In their study, Johansson et al. (2020) evaluated the differences in the assessment of sarcopenia in upper-body and lower-body strength using hand force measurement and leg force measurement, respectively. An overlap of both cut-off values for probable sarcopenia could only be detected in 4.3% of the participants. A consideration of these controversial results should also be focused on in further studies [29]. A fundamental problem of sarcopenia prevalence is the change in cut-off values published in 2010 and 2019. Scientific studies before EWGSOP2, which were published in 2019, cannot be referenced. This also includes the study by Reiss and colleagues (2019). The authors compared the cut-off values of EWGSOP [8] with those of EWGSOP2 [7]. The examined cohort presents a deviation of sarcopenia. In a gender comparison, about 53% (female) and 50% (male) after quantification of sarcopenia by EWGSOP cut-off values were not confirmed by EWGSOP2 [30]. It should be considered whether an individualized assessment of the sarcopenia criteria should be performed when the person moves into the nursing home. This would help to record changes in muscle strength, muscle mass and physical functioning over time and to finally manage them in a more targeted way.

These aforementioned problems emphasize the relevance for assessment on sarcopenia prevalence and physical functional capacity of nursing home residents. The BaSAlt study will investigate setting-specific sarcopenia prevalence. On the basis of this pilot study, adapted feasible methods for the detection and quantification of sarcopenia will be used. The project will focus on the future necessity of exercise and physical activity and their influence on sarcopenia and associated physical functional limitations.

## 5. Conclusions

This pilot study demonstrates the need to determine the prevalence of sarcopenia in German nursing homes. The quantification of sarcopenia and its numerous screening methods have shown some limitations when applied to residents in these settings. Accordingly, a practicable application of the assessments must be sought. The findings of this pilot study should lead to a feasible implementation of suitable geriatric assessments in the further investigation by the BaSAlt project.

## Figures and Tables

**Figure 1 geriatrics-06-00069-f001:**
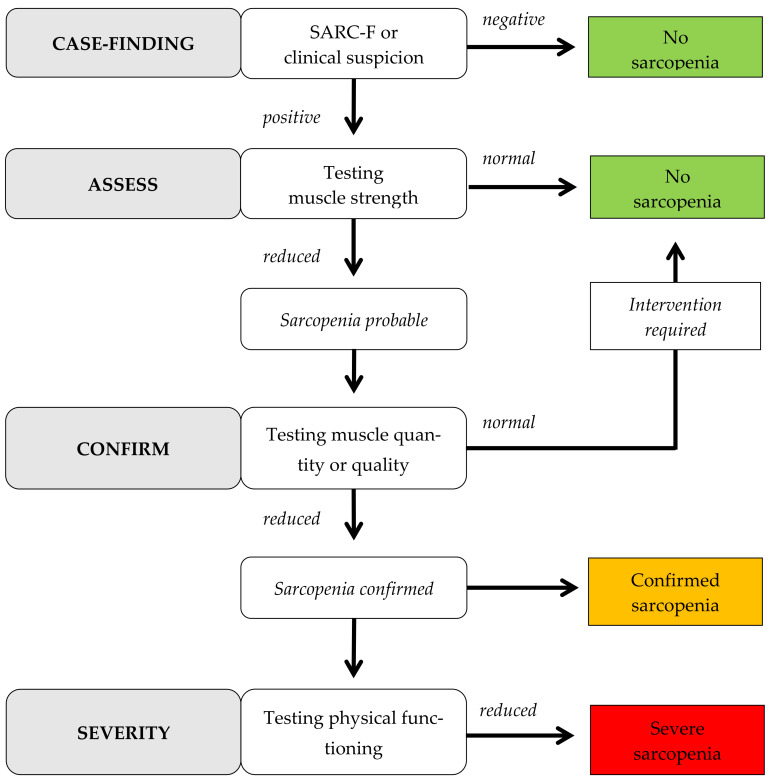
Algorithm for quantification of sarcopenia according to EWGSOP2 (modified after [7]).

**Table 1 geriatrics-06-00069-t001:** Overview of the geriatric assessment in the BaSAlt project.

Assessment	Basic Data
Demographic data	Age, sex, degree of care (inspection of patient files)
MMSE [15]	Recording of cognitive function
BI [16]	Recording of activities of daily living
Morbidity status	Recording of diseases in categories (inspection of patient files)
**Assessment**	**Primary Outcomes**
SARC-F [17]	Questionnaire for subjective self-assessment to determine sarcopenia
HFM [18]	Measurement of maximum hand force by isometric test (in kg)
BIA [19]	Measurement of ASMM (in kg) and SMI (in kg/m^2^) with BIA
4MWST (SPPB) [20]	Measurement of walking speed over a 4-meter walking distance (in m/s)
**Assessment**	**Secondary Outcomes**
Balance test (SPPB) [20,21]	Measurement of standing time in three different stand variants (in s) and CoP measurement with a Kistler^®^ force plate (in mm/m/s)
5CRT (SPPB) [20]	Measurement of strength capacity of lower extremity by repetition method (in s)
TUGsingle task [23]	Recording risk of falls, gait and balance performance (in s)
TUGmotor imagery [24]	
TUGdual task [25]	

**Table 2 geriatrics-06-00069-t002:** Cut-off values of individual assessments according to EWGSOP2 (modified after [7]).

Criteria of Sarcopenia	Cut-off-Value ♀	Cut-off-Value ♂
1. Analysis of reduced muscle strength		
HFM	<16 kg	<27 kg
5CRT	>15 s	
2. Analysis of reduced muscle quantity or quality		
ASMM	<15 kg	<20 kg
SMI	<5.5 kg/m^2^	<7.0 kg/m^2^
3. Analysis of reduced physical functioning		
4MWST	≤0.8 m/s	
SPPB	≤8 points	
TUG_single task_	≥20 s	

♀: female; ♂: male

**Table 3 geriatrics-06-00069-t003:** Demographic, physical, cognitive and clinical data of nursing home residents.

ID	Age	Walking Aids	Degree of Care	MMSE	BI	Morbidities
P1	92	rollator	2.	25	75	none
P2	88	rollator	2.	26	85	2|11
P3	91	rollator	3.	26	65	2|4|5|8|10b+
P4	77	none	3.	14	100	2|8|10b−|11
P5	88	rollator	3.	12	65	2|4|7|10b−|11
P6	84	rollator	3.	22	80	1|2|3|4|6|9II
P7	86	rollator	4.	28	80	1|8
P8	81	rollator	4.	n.p.	n.p.	1|2|3|4|5|6|7|8|9II|10b+|11
P9	91	rollator	2.	n.p.	n.p.	2|4|7|8
P10	97	none	2.	24	80	2|3

Categories of morbidities: 1, past cardiovascular events; 2, arterial hypertension; 3, coronary heart disease; 4, cardiac insufficiency; 5, cardiac pacemaker; 6, post-stroke/cerebral hemorrhage/TIA; 7, chronic lung disease; 8, cancer; 9, Diabetes Mellitus I. (I) or II. (II); 10, osteoarthritis of upper (a, or lower; b, extremities with ‘+’ or without ‘−‘ TEP.); 11, psychological/emotional/nervous disease; n.p.: assessment was not possible; n.a.; two field trials were noted in the assessment and were marked as not applicable.

**Table 4 geriatrics-06-00069-t004:** Primary outcomes.

ID	BMI	SARC-F	HFM ^1^	ASMM|SMI	4MWT Habitual|Maximum
P1♀	18.0 kg/m^2^	4/10 *	12 kg *	8.80 kg *|4.9 kg/m^2^ *	0.60 m/s *|0.62 m/s *
P2♀	32.0 kg/m^2^	6/10 *	16 kg	16.40 kg|8.2 kg/m^2^	0.47 m/s *|0.59 m/s *
P3♀	29.1 kg/m^2^	4/10 *	18 kg	21.10 kg|8.9 kg/m^2^	0.44 m/s *|0.53 m/s *
P4♀	16.1 kg/m^2^	2/10	18 kg	10.01 kg *|5.3 kg/m^2^ *	0.43 m/s *|0.67 m/s *
P5♀	25.4 kg/m^2^	7/10 *	15 kg *	n.p.|n.p.	0.38 m/s *|0.58 m/s *
P6♂	26.1 kg/m^2^	10/10 *	30 kg	20.40 kg|8.8 kg/m^2^	0.32 m/s *|n.a.
P7♀	34.9 kg/m^2^	6/10 *	18 kg	17.30 kg|7.6 kg/m^2^	0.47 m/s *|0.55 m/s *
P8♂	22.5 kg/m^2^	5/10 *	29 kg	19.80 kg *|8.2 kg/m^2^	0.22 m/s *|n.a.
P9♀	27.0 kg/m^2^	n.p.	12 kg *	12.10 kg *|7.3 kg/m^2^	0.59 m/s *|0.54 m/s *
P10♀	27.4 kg/m^2^	1/10	20 kg	15.40 kg|5.7 kg/m^2^	0.40 m/s *|0.81 m/s

♀: female; ♂: male; ^1^: values of maximum achieved hand force of the dominant hand in two passes; *: reaching cut-off values (gender-specific) according to EWGSOP2 of the respective individual assessments, SARC-F ≥4/10 points (for both sexes), HFM <16 kg **♀**/<27 kg **♂** ASMM|SMI <15 kg|< 5.5 kg/m^2^ **♀**/<20 kg|<7.0 kg/m^2^ **♂** 4MWT ≤0.8 m/s (for both sexes); n.p.: assessment was not possible; n.a.: two field trials were noted in the assessment and were marked as not applicable.

**Table 5 geriatrics-06-00069-t005:** Secondary outcomes.

ID	5CRT	SPPB	TUG_single task_	TUG_motor imagery_ ^1^	TUG_dual task_ ^1^	CoP ^1,2^
P1**♀**	n.p.	n.p.	21.19 s *	6.50 s	n.p.	n.p.
P2**♀**	n.p.	n.p.	n.p.	n.p.	n.p.	n.p.
P3**♀**	n.p.	n.p.	n.p.	n.p.	n.p.	n.p.
P4**♀**	19.78 *	8/12 *	15.41 s	10.60 s	17.31 s	119.3 mm (0.01 m/s) *^parallel^* 317.1 mm (0.03 m/s) *^semi-tandem^* 434.3 mm (0.04 m/s) *^tandem^*
P5**♀**	n.p.	n.p.	42.94 s *	n.a.	n.p.	n.p.
P6**♂**	n.p.	n.p.	n.p.	n.p.	n.p.	n.p.
P7**♀**	n.p.	n.p.	n.p.	n.p.	n.p.	n.p.
P8**♂**	n.p.	n.p.	n.p.	n.p.	n.p.	n.p.
P9**♀**	n.p.	n.p.	n.p.	n.p.	n.p.	n.p.
P10**♀**	n.p.	n.p.	n.p.	n.p.	n.p.	n.p.

♀: female; ♂: male; ^1^: individual assessments were not used by EWGSOP2 to quantify sarcopenia and were intended to further investigate physical functioning; ^2^: measurement of CoP in path length and velocity of sway using a Kistler^®^ force plate, best values for ^parallel^ (parallel stand), ^semi-tandem^ (semi-tandem stand) and ^tandem^ (tandem stand) used for evaluation; *: reaching cut-off values (gender-specific) according to EWGSOP2 of the respective individual assessments, 5CRT >15 s (for both sexes), SPPB ≥8/12 points (for both sexes); _TUGsingle task_ ≥20 s (for both sexes); n.p.: assessment was not possible; n.a.: two field trials were noted in the assessment and were marked as not applicable.

**Table 6 geriatrics-06-00069-t006:** Quantification of sarcopenia by individual case.

ID	Suspicion of Sarcopenia	Assessing the Suspicion	Confirmation of Sarcopenia	Physical Functioning	Severity of Sarcopenia
P1**♀**	positive	probably	positive	limited	severe
P2**♀**	positive	improbably	negative	limited	none
P3**♀**	positive	improbably	negative	limited	none
P4**♀**	negative	uncertain	positive	uncertain	uncertain
P5**♀**	positive	probably	not feasible	limited	uncertain
P6**♂**	positive	improbably	negative	limited	none
P7**♀**	positive	improbably	negative	limited	none
P8**♂**	not feasible	improbably	uncertain	limited	uncertain
P9**♀**	positive	probably	uncertain	limited	uncertain
P10**♀**	negative	improbably	negative	unlimited	none

**♀:** female; **♂:** male.

## Data Availability

Analyzed data sets can be requested from the corresponding author (with justification).

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
