# Peer review of "Feasibility of a Geriatric Assessment to Detect and Quantify Sarcopenia and Physical Functioning in German Nursing Home Residents—A Pilot Study"

_geriatrics, 2021, doi:10.3390/geriatrics6030069_

Round 1
Reviewer 1 Report
Even though the study is a small scale probing pilot study that is hypothesis generating, I recognize that the argument elicited by the authors are very important. Generally the paper is well written.
Major point.
Indeed, I think that EWGSOP2 attempted way too far operationalization in determining sarcopenia, while many performance test measures are apparently not possible in severely frail people. For example, individuals with higher frailty level (i.e, clinical frailty scale of 6, 7) cannot perform gait speed test or sit to stand test. However, researchers also can classify individuals unable to to these test as gait speed of 0, and sit to stand test time of infinite, in real world studies, and then sarcopenia can be diagnosed via muscle mass exam. I suggest authors to include this point in the description.
Author Response
Dear colleague, we would like to thank you for the feedback and comments.
Point 1: Indeed, I think that EWGSOP2 attempted way too far operationalization in determining sarcopenia, while many performance test measures are apparently not possible in severely frail people. For example, individuals with higher frailty level (i.e, clinical frailty scale of 6, 7) cannot perform gait speed test or sit to stand test. However, researchers also can classify individuals unable to to these test as gait speed of 0, and sit to stand test time of infinite, in real world studies, and then sarcopenia can be diagnosed via muscle mass exam. I suggest authors to include this point in the description.
Response 1: We will consider this point and include the comment in the discussion.
The requirements of EWGSOP2 by testing methods for physical functioning are too demanding for residents in nursing homes. It should therefore be considered whether physical functioning should be required as a relevant test for the quantification of sarcopenia. The need for care of the residents is also shown by the accompanying physical functional limitations. Testing of muscle strength and muscle mass can already identify a confirmed sarcopenia. Severe sarcopenia could be considered if the physical function tests are not feasible.
Reviewer 2 Report
It is a significant manuscript, worth of publishing. However I have a following comments:
- The quality of the manuscript would improve with the addition of a specific exclusion criteria section
- It would be helpful if a conclusion paragraph was added
- I detected some mistakes in the citation ex. line 275
- I minor improvement on the use of the language would be beneficial
Author Response
Dear colleague, we would like to thank you for the feedback and comments.
Point 1: The quality of the manuscript would improve with the addition of a specific exclusion criteria section.
Response 1: We will add a separate section 2.2 Inclusion and exclusion criteria to improve clarity.
The inclusion criteria for residents were determined by a degree of care ≤ 4 (classification in German care system in degree of care 1 to 5) and the ability to walk with or without walking aids. Participation was voluntary and could be interrupted either before or at any time during the assessments. Residents with a degree of care 5 could not be included due to their bedriddenness and lack of other physical or mental abilities. Residents who did not speak German language could not be included, too.
Point 2: It would be helpful if a conclusion paragraph was added.
Response 2: We will add a separate section 5. Conclusion to improve the significance of this work.
This pilot study demonstrates the need to determine the prevalence of sarcopenia in German nursing homes. The quantification of sarcopenia and its numerous screening methods have shown some limitations when applied to residents in these settings. Accordingly, a practicable application of the assessments must be sought. The findings of this pilot study should lead to a feasible implementation of suitable geriatric assessments in the further investigation by the BaSAlt project.
Point 3: I detected some mistakes in the citation ex. line 275.
Response 3: We will recheck the references and citations.
Point 4: I minor improvement on the use of the language would be beneficial.
Response 4: We will submit the revised manuscript to two native speakers for review.